# Study Protocol on the Validation of the Quality of Sleep Data from Xiaomi Domestic Wristbands

**DOI:** 10.3390/ijerph18031106

**Published:** 2021-01-27

**Authors:** Patricia Concheiro-Moscoso, Francisco José Martínez-Martínez, María del Carmen Miranda-Duro, Thais Pousada, Laura Nieto-Riveiro, Betania Groba, Francisco Javier Mejuto-Muiño, Javier Pereira

**Affiliations:** 1CITIC, TALIONIS Group, Elviña Campus, Universidade da Coruña (University of A Coruña), 15071 A Coruña, Spain; patricia.concheiro@udc.es (P.C.-M.); f.martinezm@udc.es (F.J.M.-M.); carmen.miranda@udc.es (M.d.C.M.-D.); thais.pousada.garcia@udc.es (T.P.); laura.nieto@udc.es (L.N.-R.); javier.pereira@udc.es (J.P.); 2Faculty of Health Sciences, Oza Campus, Universidade da Coruña (University of A Coruña), 15071 A Coruña, Spain; 3Clinical Neurophysiology Service, Hospital San Rafael, 15009 A Coruña, Spain; fmejmui@gmail.com

**Keywords:** sleep, health promotion, daily life activities, occupation, polysomnography, Xiaomi Mi Smart Band 5, wearable technology, participatory health, internet of things

## Abstract

(1) *Background*: Sleep disorders are a common problem for public health since they are considered potential triggers and predictors of some mental and physical diseases. Evaluating the sleep quality of a person may be a first step to prevent further health issues that diminish their independence and quality of life. Polysomnography (PSG) is the “gold standard” for sleep studies, but this technique presents some drawbacks. Thus, this study intends to assess the capability of the new Xiaomi Mi Smart Band 5 to be used as a tool for sleep self-assessment. (2) *Methods*: This study will be an observational and prospective study set at the sleep unit of a hospital in A Coruña, Spain. Forty-three participants who meet the inclusion criteria will be asked to participate. Specific statistical methods will be used to analyze the data collected using the Xiaomi Mi Smart Band 5 and PSG. (3) *Discussion*: This study offers a promising approach to assess whether the Xiaomi Mi Smart Band 5 correctly records our sleep. Even though these devices are not expected to replace PSG, they may be used as an initial evaluation tool for users to manage their own sleep quality and, if necessary, consult a health professional. Further, the device may help users make simple changes to their habits to improve other health issues as well. Trial registration: NCT04568408 (Registered 23 September 2020).

## 1. Introduction

Sleep is an occupational area that has considerable implications on our daily life [1,2]. Thus, sleep disorders have become one of the most important common problems in public health [3]. The prevalence of sleep disorders increases with age [4], even though they can appear at any life stage. Bad quality and quantity of sleep, sleep arousals, and a strong will to take diurnal naps are the main complaints among the general population [5,6]. Epidemiological studies indicate that between 20% and 48% of adults between 34 and 60 years old have difficulties initiating and maintaining sleep [7]. In fact, insomnia, which is the most common sleep disorder, is present in 30–45% of the population [8].

Several studies state that sleep disorders are related to health status and quality of life in the general population [9], leading to difficulties in the performance of daily life activities, which in turn causes physical exhaustion, low productivity, a greater risk of falling, mood problems, and diurnal sleep, among many others [10,11,12,13,14].

These factors related to sleep disorders cause a direct and indirect economic burden, causing public health costs to rise [15,16]. Different studies have shown a direct relation between sleep disorders and costs in primary care and hospitals. Lee et al. indicated that the population suffering from sleep disorders regularly attends the emergency room to consult professional health workers, or demands telematic assistance [9]. This is due to subjective complaints of sleep, along with factors related to stress, working pressure, and other chronic diseases [17,18].

Diverse sleep disorders are diagnosed using different tests that assess the quality and quantity of sleep [19,20]. Objective and valid diagnostic tests that evaluate the quality and quantity of sleep are a previous requisite for the control of sleep disorders [21]. Polysomnography (PSG) is considered by the scientific community as the most reliable test for the measurement of sleep parameters [22,23]. However, it has some clear drawbacks, such as its costs, operation, invasiveness, and time needed for its use [24,25]. To address these problems, actigraphy technology was designed. Actigraphy consists of the measure of sleep parameters using a wearable device created for clinical use [26]. Several studies have found similar measures from both actigraphy and PSG, proving its fitness for these studies [27,28,29]. Although actigraphy has become a clinically useful tool to test sleep disorders, it cannot provide feedback to people because it purpose is mainly of a research nature [26]. Thus, inspired by the actigraphy device, researchers have recently focused on the validation of different tools for sleep self-assessment, such as activity wristbands [30].

In recent years, activity wristbands have become widely accepted among the population due to their low price, easy use, smart design, and feedback that they offer to users through their fast sync with smartphones, boosting participatory health [31,32,33]. These devices have already reported benefits in clinics, as they reduce care burdens and facilitate the early diagnosis of health disorders [34]. They also allow the general population to acquire information on their sleep quality and physical activity, which may help in the early detection or prevention of symptoms related to sleep alterations [35,36,37].

Despite being limited, some studies highlight the importance of the validation of these devices not only from a technological point of view, but also for health promotion, since, by allowing sleep self-assessment, they are tightly related to improvements in life quality and daily functioning [38,39,40,41].

Recent studies have compared the measurements of these wristbands with those of actigraphy and PSG techniques. PSG involves different types of electrodes (electroencephalogram (EEG), electrooculogram (EOG), electromyogram (EMG), and electrocardiogram (ECG)) to classify the stages of sleep [42]. A few studies have shown that the usability of an ECG signal can determine the stages of sleep [43,44,45,46]. Nevertheless, some have considered the need for further research due to the insufficient quality of this classification compared to the combination of the other electrodes [47,48].

In general, wearable devices can determine the stages of sleep through a heart rate (HR) sensor, which measures the HR, and an accelerometer that detects movement. Previously validated wearable devices have shown a high precision and sensitivity, but also a low specificity and poor agreement in sleep-stage classification when compared with PSG [20,32,33,49,50]. It would be logical to expect these devices to measure sleep more accurately; however, according to these studies, there is a wide range of improvements that can still be accomplished [51,52]. As a result, this protocol has been designed to validate the Xiaomi Mi Smart Band 5 due to its low cost and high acceptance among users, as it is the best-selling fitness tracker [53]. This device has the capability of tracking the activity and HR of users, and classifies the user’s state as being awake or in light sleep, deep sleep, or REM sleep [54].

Therefore, this research focuses on determining whether sleep stages recorded by the new Xiaomi Mi Band 5 can effectively replace PSG sleep-stage classification in sleep-study participants. To this end, the following features will be calculated for both devices: total sleep time, sleep efficiency, sleep latency, and wake after sleep onset. Sleep stages will also be recorded.

Thus, the primary purpose of the present study is to validate the quality of the data generated by Xiaomi wearables when compared with the PSG technique from a hospital sleep unit.

The secondary objectives are: (1) to determine the total sleep time, sleep efficiency, sleep latency, and wake after sleep onset from both the wearable device and PSG; and (2) to examine whether the classification of sleep stages provided by wearable devices is comparable with classifications based on PSG.

## 2. Materials and Methods 

### 2.1. Study Design

This is a pilot study that aims “to demonstrate that the expected measures, data collecting instruments, and their management system are viable and effective” [55]. This is an observational and prospective study, which means that different variables from the population of interest will be observed and recorded without any direct intervention, in order to establish causality associations between these variables. It will be considered a longitudinal study, with a timeframe of six months, in which participants will stay one night at the sleep unit. They will be monitored to record their sleep through PSG and the Xiaomi wristband.

This study protocol follows the SPIRIT 2013 checklist for study protocols for clinical trials (See Appendix A) [56].

### 2.2. Study Settings

The setting of this study will be the sleep unit of a hospital in A Coruña, Spain. This service includes a special consulting room where different sleep disorders are diagnosed and treated. Some of these disorders are sleep apnea, insomnia, restless-legs syndrome, narcolepsy, the delay and advance of sleep stages, sleep terrors, somnambulism, and bruxism. The sleep laboratory is equipped to perform different diagnostic tests: nocturnal and diurnal PSG, multiple latency tests, immobilization tests, and wakefulness maintenance tests.

### 2.3. Eligibility Criteria

Sample selection will be performed through an intentional sampling based on inclusion and exclusion criteria.

Participant inclusion criteria: (a) having attended the sleep unit to have any diagnostic test done; (b) having declared an age that is equal to or higher than 18 years old.

Participant exclusion criteria: (a) having a significant health complication that hinders active participation in the study; (b) suffering from skin hypersensitivity or an allergic reaction due to the materials of the wristbands that will be used in the study.

### 2.4. Recruitment Process

This project will be conducted in the sleep unit of a hospital that, regardless of this study, people attend to have PSG tests done with the aim of detecting possible alterations in their sleep. The participants will be assisted by, at a minimum, the clinical neurophysiologist responsible for the sleep unit and a member of the research team. In the first meeting with each of the users, the clinical neurophysiologist responsible for recruiting the participants and monitoring them will inform them, if they meet the selection criteria, about the possibility of participating in the study and the implications of their participation, i.e., the use of an activity bracelet during the day or night and what is involved in the PSG test that will be performed on them. In addition, the mechanisms that will be followed to guarantee their anonymity and the confidentiality of their data will be clearly stated to the participants.

After the main characteristics of the research have been explained, each potential participant will be given an information sheet so that they can consult the information and make a decision before the test is carried out. Once the users return to the unit for testing, any doubts or queries will be resolved by the clinical neurophysiologist, and the informed consent document will be signed, if applicable, by the responsible professional and the person who expresses their final interest in participating in the study. In addition, if the participant has reading and writing difficulties, a witness must be present during the entire procedure to confirm that all the ethical processes have been respected by the research team. On the day of the PSG test, participants will arrive three hours before the test to become acquainted with the room, and will turn the lights on and off based on their sleep pattern. The specialized technical staff will control the test and will attend to any possible patient demands. Figure 1 shows the recruitment and the assessment process.

### 2.5. Justification of Sample Size

Differences equal or greater than 15 min in deep-sleep time between the measures of both methods are considered clinically relevant. Accepting a 0.05 alpha risk and a 0.1 beta risk (90% power) in a bilateral contrast, 43 subjects are needed to detect a difference that is equal to or greater than 15 min. Conservatively, a standard deviation of 30 for mean differences is assumed, according to previous studies [21,50,57].

### 2.6. Outcomes

The outcomes will be the sleep-stage identification of PSG and the Xiaomi Mi Smart Band 5, and measurements of total sleep time, sleep efficiency, sleep latency, and wake after sleep onset.

### 2.7. Data Collection and Management

The study will focus on examining the following variables (Table 1): time in bed, hours of light, deep and Rapid-Eye-Movement (REM) sleep, sleep efficiency, sleep latency, wake after sleep onset, and HR. Additionally, the Pittsburgh Sleep Quality Index (PSQI) scale and a sociodemographic questionnaire will be used to assess participants’ self-awareness of their sleep.

The sociodemographic questionnaire will collect different data: gender, year of birth, weight, height, handedness, sleep pathologies, and the assessment test. These data will be pseudonymized and transferred by the hospital to the research group with the prior consent of the participant.

The PSQI will evaluate subjective sleep quality. This questionnaire consists of 24 items about the area of sleep. This tool analyzes the quantity, quality, duration, latency, and efficiency of sleep [58].

The objective sleep data will be obtained through two different devices:

Xiaomi Mi Smart Band 5: This is a wearable device focused on health and sport that measures biomedical parameters of users, among which HR and sleep measurements are the most remarkable. The wristband also requires certain personal data from the user to precisely calculate the activity, such as age, height, weight, gender, handedness, and wristband location. Wristband data will be obtained from the Mi Fit application, which is the native app of the Xiaomi Mi Band wristband. It is necessary to connect the wristband via Bluetooth to a smartphone of general use for the group, creating generic emails for each of the wristbands. The features of interest will be calculated from the data shown in this app [32].

PSG: Data obtained from diurnal and nocturnal PSG will be collected, as this is the standard test for patients who attend to the hospital’s sleep unit. PSG measures sleep cycles and stages (N1, N2, N3, and REM sleep) by recording different variables, such as brain waves, eye movement, skeletal muscle activity, heart frequency and rhythm, blood pressure, oxygen level in the blood, breathing patterns, body position, limb movements, and whether there is snoring or other noise [59]. PSG parameters will be scored in 30 s epochs according to the American Academy of Sleep Medicine (AASM) guidelines [60], and PSG recordings will be exported in the European Data Format+ format [61] and interpreted by a doctor specialized in the study of sleep. Thus, sleep-stage classification based on PSG will be considered the “gold standard” for this study.

The data will be pseudonymized at the moment they are recorded; thus, the confidentiality of the data collected and the anonymity of each participant will be maintained. Once the project has ended, each participant’s data will be stored for future studies if they provide consent.

### 2.8. Data Analysis

The analysis will be performed by using the statistical software R. Numeric variables will be expressed as mean (M) and standard deviation (SD), including the range, minima, and maxima. Beyond simple data and study variable descriptions, inferential analysis will be performed in order to determine possible significant relationships between the variables of the study.

A paired-sample Student’s t-test will be used to compare the means of the different sleep parameters of interest. T-test effect sizes are 0.2 (small effect), 0.5 (moderate effect), and 0.8 (large effect), so if the means of two groups do not differ by a 0.2 standard deviation, the difference is trivial, even if there is a statistically significant relationship. [62]. Bland–Altman plots will be used to assess the concordance between both devices for each of the sleep parameters (total sleep time, sleep efficiency, and wake after sleep onset). A positive bias indicates that the device tends to underestimate a variable when compared with PSG. A negative bias indicates that a sleep variable is overestimated [63]. Point estimations will be calculated, as well as their 95% confidence interval.

An epoch-by-epoch (EBE) (min × min) analysis will be performed in order to calculate the sensitivity (the proportion of epoch segments identified as a sleep state by the PSG that are correctly identified by Xiaomi), specificity (the proportion of epoch segments identified as a waking state by the PSG that are correctly identified by the Xiaomi wristband), and the level of agreement between both devices for light sleep identification (the proportion of PSG-classified N1 + N2 stages identified as light sleep by the Xiaomi wristband), deep sleep identification (the proportion of PSG N3 + N4 stages identified as deep sleep by the Xiaomi wristband), and REM sleep identification (proportion of PSG-classified REM stages identified as REM sleep by the Xiaomi wristband) [64]. Data analysis will include the cleaning or preprocessing, description, and processing of the stored data. The final aim of this workflow will be to obtain useful information that can be used for decision-making. During preprocessing, the wrong values of the dataset will be removed or corrected to avoid bias in the results. Subsequently, obtaining a descriptive statistic study that summarizes relations and distributions in a simple way will help us to know which processing strategy needs to be taken. When processing, information extraction will be performed.

### 2.9. Ethics and Dissemination

This study protocol was approved by the A Coruña-Ferrol Research Ethics Committee, under the number 2020/318, on 20 July 2020. In addition, this protocol was registered in Clinical Trials Protocol Registration and Results system on 23 September 2020, available at https://clinicaltrials.gov/ct2/show/NCT04568408. In case any change in the protocol is needed, this will be communicated to the ethics committee with the assigned reference number. These modifications will also be updated in the clinical trials registry.

For each participant, the process of informed consent will be applied. Participants will receive complete verbal and written information about the characteristics of the study and about the implications derived from their participation in it. An information sheet that they can read slowly will be given to each participant, and they will be able to ask any questions they may have. Once it has been ensured that all participants fully understand the information provided, they will decide on whether or not they wish to participate in the study and, if they decide to participate, agree by signing the informed-consent document. The main researcher will maintain the confidentiality of all data collected and the anonymity of each participant. Thus, the Spanish and European Organic Law on the protection of personal data will be respected at all times [65,66]. The data of the participants will be collected, encoded, and preserved until the end of the study. Once the study is finished, each participant’s data will be stored for future studies if they provide consent. The results and conclusions of this study will be disseminated through their publication in influential scientific journals and their presentation at appropriate conferences.

## 3. Discussion

The validation conducted in this study will allow us to evaluate whether the Xiaomi Mi Smart Band 5 is a device that can be used as a health promoter by allowing sleep self-assessment in the general population. Due to its low price, it is fairly accessible to any user who considers that they may be suffering from a sleep disorder that is potentially worsening their health [32]. Hence, if the data generated by the smart band proves to be reliable, any health professional could evaluate at a glance whether the user is suffering from a sleep disorder, and whether it influences their daily life activities [40,67]. The widespread use of these devices could help users to self-assess their sleep and connect with health professionals in the case of bad sleep quality, which would imply a better prognosis and a longer and higher-quality life expectancy. It is important to highlight some limitations and risks in studies focused on the validation of wearables, since firmware updates may modify the sleep-tracking algorithm, and the intra-wearables’ sleep measurements do not perfectly match, even when multiple smart bands worn by the same user are identical.

## 4. Conclusions

This project contributes to the validation of the sleep data obtained through the Xiaomi MiBand 5, making a comparison with the PSG data. Therefore, it is intended to determine whether the sleep parameters (TIB, SOL, WASO, SE, light sleep, deep sleep and REM sleep) are correctly measured through the Xiaomi MiBand 5.

The widespread use of these devices by the current society is not expected to replace the PSG. However, the information obtained by the wristbands about sleep is user-friendly in a domestic context, as this sleep data doesn’t need to be processed as in the case of the PSG. Thus, the use of these devices can help the population to be able to objectively determine whether they have difficulties in their sleep or not, as well as to promote healthy lifestyle habits, and if necessary, seek the advice of a healthcare professional.

## Figures and Tables

**Figure 1 ijerph-18-01106-f001:**
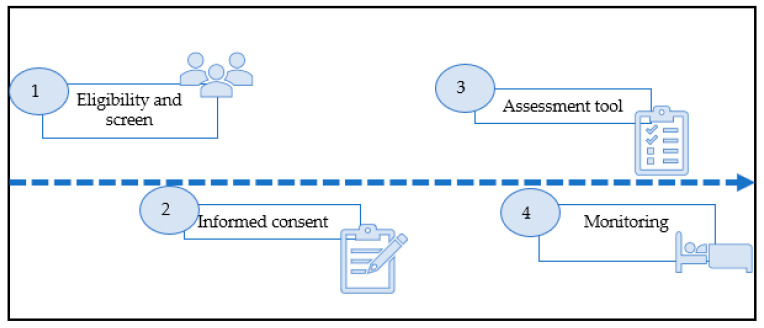
The recruitment and assessment process.

**Table 1 ijerph-18-01106-t001:** Summary of the features of interest for our study.

Variable	Description	Dimension
Time in bed (TIB)	Total time the patient is laying down	min
Sleep onset latency (SOL)	Length of time from full wakefulness to sleep	min
Wake after sleep onset (WASO)	Periods of wakefulness after defined sleep onset	min
Sleep efficiency (SE)	Time asleep/TIB × 100	%
Light sleep	N1 + N2 sleep stages	min
Deep sleep	N3 sleep stages	min
Rapid Eye Movement (REM) sleep	-	min

## Data Availability

Once the data collection process is finished and these have been coded, structured, and analyzed, these data will be provided, provided that the Spanish Data Protection Agency consents, to any researcher who contacts the TALIONIS group’s Principal Investigator, Javier Pereira (Javier.pereira@udc.es).

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
