# Peer review of "Study Protocol on the Validation of the Quality of Sleep Data from Xiaomi Domestic Wristbands"

_ijerph, 2021, doi:10.3390/ijerph18031106_

Round 1

Reviewer 1 Report

Wearable devices enable intelligent interactions between humans and computers and provide new services to users with the help of artificial intelligence. The paper presents a validation study of the sleep data collected from wearable device Xiaomi domestic wristbands and PSG.

The data collection and preprocessing process are fine. The paper is lack of data analytics, even it has been pointed out that statistics methods and software R can be used for analysis. No results and conclusions have been presented to validate whether wearable Xiaomi domestic wristbands can be used for the sleep quality study and to what extend. There are interested parameters and would be good to point the performance of each interested parameter between Xiaomi and PSG to validate the study.

Author Response

Dear reviewer,

We thank you for your valuable and useful contribution which are of great help to the improvement of our manuscript. We attach the responses to your comments.

Point 1: Wearable devices enable intelligent interactions between humans and computers and provide new services to users with the help of artificial intelligence. The paper presents a validation study of the sleep data collected from wearable device Xiaomi domestic wristbands and PSG.

The data collection and preprocessing process are fine. The paper is lack of data analytics, even it has been pointed out that statistics methods and software R can be used for analysis. No results and conclusions have been presented to validate whether wearable Xiaomi domestic wristbands can be used for the sleep quality study and to what extend. There are interested parameters and would be good to point the performance of each interested parameter between Xiaomi and PSG to validate the study.

Response1: Thank you for your suggestions and feedback. It is true that the study does not present results or conclusions. However, this study is a protocol that aims to validate a wearable device. Furthermore, SPIRIT guidelines are followed, which state that a protocol is "a document that provides sufficient detail to enable understanding of the background, rationale, objectives, study population, interventions, methods, statistical analyses, ethical considerations, dissemination plans, and administration of the trial; replication of key aspects of trial methods and conduct; and appraisal of the trial's scientific and ethical rigor from ethics approval to the dissemination of results” [1]. Thus, the performance of the parameters between Xiaomi and the PSG is a possible result of the study, and as well as the other results associated with the validation, they will be presented once the recruitment of the study has been completed in a future publication.

In addition, the authors used the IJERPH journal service of English Editing to ensure the quality of the writing. These changes are marked in yellow along the manuscript. The authors attach the certificate of the manuscript's English editing.

Reference: 1. Chan AW, Tetzlaff JM, Altman DG, Laupacis A, Gøtzsche PC, Krleža-Jerić K, et al. SPIRIT 2013 statement: defining standard protocol items for clinical trials. Ann Intern Med. 2013;158(3):200–7.

Reviewer 2 Report

Thank you for giving me the opportunity to review the interesting study protocol. Enthusiasm for the manuscript is reduced due to several identified major and minor concerns which if addressed could strengthen the manuscript.

Major concern

  1. It is a quite interesting design to compare between the PSG and Xiaomi Mi Smart Band 5 for the practical realization. As you know, there are some complex criteria for classifying sleep stage by PSG. For example, PSG is based on EEG, EOG, EMG, and so on. However, how can you assess the Xiaomi Mi Smart Band 5? Using ECG (HR)? If yes, the authors should mention the correlation between ECG and PSG outcome or possibilities that ECG can be used for assessing sleep stage in introduction. Also, the actigraphy also could assess the sleep efficiency, total sleep time, sleep latency, and wake after sleep onset based on sleep-wake judgment for their algorithm (Actigraphy could not assess the REM, and light and/or deep sleep stage).
  2. Is this study design a cross-sectional study, observational study, or longitudinal study? There is some confusing statement in L97-105. Please modify it.
  3. Also, the experiment using PSG generally consider the adaptation night to avoid first day night effects. It is a quite important argument in this study. Please consider it.
  4. How is the PSG parameter judged? Is this according to AASM or anything else? Please add it.
  5. If you use t-test, please add the effect-size and insert reference to the method of previous studies in the L148.    

Minor concern

  1. The authors checked an item number of 2a in SPIRIT checklist. But there is no Trial Registration section. Please add the “Trial Registration” in Abstract.
  2. There are duplicate explanations of the L41-43 and L49-50. Please modify the side effect of low sleep quality.
  3. I suggest adding the limitation of the actigraphy such as “it could not feedback to participants because there are produced for the research” in wrote L64~66 before L60 [Inspired by the ~].
  4. Please modify L104. (from ‘checklist for study protocols of randomized controlled trial’ to ‘checklist for study protocols for clinical trials’)
  5. The abbreviation of REM, N1, N2, N3 should be full name as mentioned for the first time. Also, there is a need to correct Table 1 in N1, N2, and L200-201 in F1, F2, F3, F4.
  6. Please add “subjective” in L162 as followed, ‘The PSQI will evaluate subjective sleep quality’. Also, please add “objective” in L164 as followed, ‘The objective sleep data will be ~’.
  7. Please check the references in (L139, 154) before the final submission.

Author Response

Dear reviewer,

We thank you for your valuable and useful contribution which is of great help to the improvement of our manuscript. We attach the responses to your comments.

Major Concern

Point 1: It is a quite interesting design to compare between the PSG and Xiaomi Mi Smart Band 5 for the practical realization. As you know, there are some complex criteria for classifying sleep stage by PSG. For example, PSG is based on EEG, EOG, EMG, and so on. However, how can you assess the Xiaomi Mi Smart Band 5? Using ECG (HR)? If yes, the authors should mention the correlation between ECG and PSG outcome or possibilities that ECG can be used for assessing sleep stage in introduction. The actigraphy also could assess the sleep efficiency, total sleep time, sleep latency, and wake after sleep onset based on sleep-wake judgment for their algorithm (Actigraphy could not assess the REM, and light and/or deep sleep stage).

Response 1: Thank you for your suggestion and appreciation. The Xiaomi Mi Smart Band 5 detects the sleep stages using a combination of the Heart Rate (HR) and movement. These parameters are obtained by an HR sensor and an accelerometer. The authors referred to some studies that research the use of ECG for assessing the sleep stage. This information is marked in yellow on page 2, lines 76-81. In addition, the authors highlighted the sleep variables that will be extracted from Xiaomi Mi Band 5 and PSG. See it in Table 1 on page 5, line 191.

Point 2: Is this study design a cross-sectional study, observational study, or longitudinal study? There is some confusing statement in L97-105. Please modify it.

Response 2: Thank you for your appreciation. This is an observational and prospective study because the authors will use periodic observations collected predominantly following users' enrolment. And, it’s a longitudinal study because the authors collected the variables at different times. This changed is marked in yellow in lines 105-110, page 3.

Point 3: Also, the experiment using PSG generally consider the adaptation night to avoid first day night effects. It is a quite important argument in this study. Please consider it.

Response 3: Thank you for your suggestion. The authors agree with the reviewer's appreciation. However, the participants only go for one night because of the sleep unit's organization. But on the day of the PSG test, the participants come 3 hours before to get used to the room where the test will take place. The sleep technician attends to the possible participant's demands and ensures that it spends the night as comfortably as possible. This information is marked in yellow on page 4, lines 145-150.

In addition, the authors didn't consider this argument because the main objective of the study is to determine whether the Xiaomi wearable data are comparable with the PSG classification.

Point 4: How is the PSG parameter judged? Is this according to AASM or anything else? Please add it.

Response 4: Thank you for your suggestion. The authors judge the PSG parameters according to the American Academy of Sleep Medicine (AASM). You can see it marked in yellow on page 5 lines 184-185.

Point 5: If you use t-test, please add the effect-size and insert reference to the method of previous studies in the L148.

Response 5: Thank you for your comment. The authors added the effect size on page 5, lines 198-200; and insert the references of previous studies on page 4, line 156.

Minor concern

Point 1: The authors checked an item number of 2a in SPIRIT checklist. But there is no Trial Registration section. Please add the “Trial Registration” in Abstract.

Response 1: Thank you for your appreciation. The authors added the Trial registration, marked in yellow in the abstract (page 1, line 30-31).

Point 2: There are duplicate explanations of the L41-43 and L49-50. Please modify the side effect of low sleep quality.

Response 2: Thank you for your comment. The authors modified the side effects of low sleep quality. See you this change, marked in yellow, on page 2, lines 52-53.

Point 3: I suggest adding the limitation of the actigraphy such as “it could not feedback to participants because there are produced for the research” in wrote L64~66 before L60 [Inspired by the ~].

Response 3: Thank you for your suggestion. The authors added the limitation of the Actigraphy. It is marked in yellow on page 2, lines 62-63.

Point 4: Please modify L104. (from ‘checklist for study protocols of randomized controlled trial’ to ‘checklist for study protocols for clinical trials’)

Response 4: Thank you for your appreciation. The authors modified the name of the checklist. It’s marked in yellow on page 3, line 111.

Point 5: The abbreviation of REM, N1, N2, N3 should be full name as mentioned for the first time. Also, there is a need to correct Table 1 in N1, N2, and L200-201 in F1, F2, F3, F4.

Response 5: Thank you for your comment. The authors add the abbreviation of sleep’ stages on page 4, line 180-181. Also, the authors corrected the mistakes in Table 1 and lines 209-210.

Point 6: Please add “subjective” in L162 as followed, ‘The PSQI will evaluate subjective sleep quality’. Also, please add “objective” in L164 as followed, ‘The objective sleep data will be ~’.

Response 6: Thank you for your suggestion. The authors added the word “subjective”, marked in yellow, in line 168 on page 4 and, we added the word “objective”, marked in yellow, in line 170, page 4.

Point 7: Please check the references in (L139, 154) before the final submission.

Response 7: Thank you for your appreciation. The authors check the references in lines 139, 154. This reference is marked in yellow on pages 4 and 5, lines 148,161.

In addition, the authors use the IJERPH journal service of English Editing to ensure the quality of the writing. These changes are marked in yellow along with the manuscript. The authors attach the certificate of the manuscript's English editing.

Round 2

Reviewer 2 Report

This paper is an important contribution and I recommend that it be accepted for publication.